# Emerging Invasive Fungal Infections in Critically Ill Patients: Incidence, Outcomes and Prognosis Factors, a Case-Control Study

**DOI:** 10.3390/jof7050330

**Published:** 2021-04-24

**Authors:** Romaric Larcher, Laura Platon, Matthieu Amalric, Vincent Brunot, Noemie Besnard, Racim Benomar, Delphine Daubin, Patrice Ceballos, Philippe Rispail, Laurence Lachaud, Nathalie Bourgeois, Kada Klouche

**Affiliations:** 1Intensive Care Medicine Department, Lapeyronie Hospital, Montpellier University Hospital, 371, Avenue du Doyen Gaston Giraud, 34090 Montpellier, France; l-platon@chu-montpellier.fr (L.P.); amalric.matthieu@gmail.com (M.A.); v-brunot@chu-montpellier.fr (V.B.); n-besnard@chu-montpellier.fr (N.B.); r-benomar@chu-montpellier.fr (R.B.); d-daubin@chu-montpellier.fr (D.D.); k-klouche@chu-montpellier.fr (K.K.); 2PhyMedExp, INSERM (French Institute of Health and Medical Research), CNRS (French National Centre for Scientific Research), University of Montpellier, 34090 Montpellier, France; 3Hematology Department, Saint Eloi Hospital, Montpellier University Hospital, 34090 Montpellier, France; p-ceballos@chu-montpellier.fr; 4Mycology and Parasitology Laboratory, Lapeyronie Hospital, Montpellier University Hospital, 34090 Montpellier, France; p-rispail@chu-montpellier.fr (P.R.); l-lachaud@chu-montpellier.fr (L.L.); n-bourgeois@chu-montpellier.fr (N.B.); 5MiVEGEC (Infectious Diseases and Vectors: Ecology, Genetic, Evolution and Control), IRD (Research and Development Institute), CNRS, University of Montpellier, 911 Avenue Agropolis, 34394 Montpellier, France

**Keywords:** intensive care unit, outcome, invasive fungal infections, mucormycosis, *Fusarium*, *Scedosporium*, *Trichosporon*, *Saprochaete*, *Chaetomium*, *Saccharomyces*

## Abstract

Comprehensive data on emerging invasive fungal infections (EIFIs) in the critically ill are scarce. We conducted a case-control study to characterize EIFIs in patients admitted to a French medical ICU teaching hospital from 2006 to 2019. Among 6900 patients, 26 (4 per 1000) had an EIFI: *Mucorales* accounted for half, and other isolates were mainly *Saprochaete*, *Fusarium* and *Scedosporium*. EIFIs occurred mostly in patients with immunosuppression and severe critical illness. Antifungal treatments (mainly amphotericin B) were administered to almost all patients, whereas only 19% had surgery. In-ICU, mortality was high (77%) and associated with previous conditions such as hematological malignancy or cancer, malnutrition, chronic kidney disease and occurrence of acute respiratory distress syndrome and/or hepatic dysfunction. Day-90 survival rates, calculated by the Kaplan–Meier method, were similar between patients with EIFIs and a control group of patients with aspergillosis: 20%, 95% CI (9- 45) versus 18%, 95% CI (8- 45) (log-rank: *p* > 0.99). ICU management of such patients should be assessed on the basis of underlying conditions, reversibility and acute event severity rather than the mold species.

## 1. Introduction

Fungal infections are life-threatening opportunistic infections that have emerged as a major cause of morbidity and mortality in critically ill patients. Yeast infections, mainly *Candida* spp., remain much more frequent than mold infections [1]. Among mold infections, *Aspergillus* spp. represent the most frequently isolated filamentous fungi in these circumstances [1]. Though uncommon, other filamentous fungi, such as *Mucorales*, *Fusarium*, *Scedosporium* or *Saprochaete* have emerged over recent years not only in patients with hematologic malignancies or bone marrow transplants and may be underestimated [2]. These emerging invasive fungal infections (EIFIs) share common features such as low incidence rate, difficult diagnosis and treatment, multi-organ involvement ability and high morbidity and mortality rate [1,3].

Previous studies on EIFIs mostly focused on mucormycosis and did not investigate specifically critically ill patients [4,5,6]. Recently, Claustre et al. reported the clinical presentations of 74 intensive care unit (ICU) patients with mucormycosis and highlighted their poor outcome [7]. Also, they reported the critical role of previous conditions such as hematological malignancies and malnutrition [7]. However, comprehensive data about ICU patients with EIFIs remain lacking, especially in those with other species than *Mucorales*. Indeed, mortality rates and prognosis factors remain unknown in ICU patients with EIFIs such as fusariosis, scedosporiosis and trichosporonosis.

We aimed, therefore, to describe in this retrospective critically ill cohort study the epidemiological trends, clinical features and outcomes of EIFIs and to assess prognostic factors in ICU settings. We also compared our cohort of ICU patients with EIFIs to a matched control group with aspergillosis in order to evaluate the weight of these species on outcome.

## 2. Materials and Methods

### 2.1. Study Design and Settings

This observational, retrospective cohort study was conducted from January 2006 to December 2019 in the Medical ICU of Montpellier University Hospital. In our hospital, this unit is the chief site of onco-hematology and kidney transplant patient management whenever they present a severe and life-threatening condition necessitating mechanical ventilation or/and hemodynamic support.

The study was registered at ClinicalTrials.gov (No. NCT04647539) on 23 November 2020.

### 2.2. Patients

During the study period, eligible patients were screened through the local mycology laboratory database and the hospital coding records. Medical charts of patients admitted to the ICU with a positive culture for molds other than *Candida* and *Aspergillus* were independently reviewed by two ICU physicians (R.L. and L.P.) and a mycologist (Nathalie Bourgeois) to confirm and secure an EIFI diagnosis according to European Organization for Research and Treatment of Cancer (EORTC) and European Confederation of Medical Mycology (ECMM) criteria [8,9]. A proven EIFI was retained in a host with illness consistent with EIFI and mycological evidence of EIFI or a pathognomonic histopathologic or direct microscopic exam of EIFI from blood or a specimen obtained from the affected site. A probable EIFI was retained in the presence of a host factor plus a clinical picture consistent with EIFI and mycological evidence from urine, blood or CSF. A possible EIFI was retained in the presence of a host factor and a positive polymerase chain reaction (PCR) for mold species responsible for EIFI.

### 2.3. Data Collection

Epidemiological and clinical data were collected including the Charlson comorbidity index [10] and reasons for ICU admission. Malnutrition was defined as follows: body mass index (BMI) < 20 kg/m^2^, weight loss within past 6 months ≥ 5% and serum albumin < 30 g/L [11]. A simplified acute physiology score (SAPS) II and sequential related organ failure assessment (SOFA) score were calculated at admission and at EIFI diagnosis [12,13]. Risk factors for invasive fungal infection such as prior broad-spectrum antimicrobial therapy, central venous catheter, and prior immune status were collected [1,9]. ICU management, occurrence of septic shock and organ dysfunction–acute kidney injury (AKI), acute respiratory distress syndrome (ARDS), myocardial dysfunction: left ventricular ejection fraction < 50% and hepatic dysfunction: bilirubin > 20 µmol/L were recorded [13,14,15,16]. Mold species were identified using morphologic characteristics, matrix-assisted laser desorption/ionization time-of-flight (MALDITOF) mass spectrometry or genetic sequencing. Cultures of isolates and antifungal susceptibility tests were performed according European Committee on Antimicrobial Susceptibility Testing (EUCAST) guidelines. We also collected antifungal therapies and time elapsed from ICU admission to EIFI diagnosis as well as from EIFI diagnosis to adequate antifungal therapy. Adequacy of antifungal therapies were retrospectively confirmed by two investigators (R. L. and L.P.) in agreement with recently published international guidelines [9,17,18,19,20]. Consequently, antifungal therapies that were considered adequate at management date may have been judged inadequate according to recent guidelines. Survival at ICU discharge and day 90 and date of death, if applicable, were recorded. We then identified predictive factors of in-ICU mortality.

We compared clinical data and the prognosis of this cohort to that of a control group of patients with aspergillosis in order to best evaluate the impact of EIFIs in the critically ill. Controls were selected among patients admitted to our ICU during the same period with a diagnosis of aspergillosis according to Infectious Diseases Society of America (IDSA) guidelines [21]. Controls were then matched based on age, SOFA score, development of ARDS and previous conditions (at least 3 similar conditions among hemopathy/cancer, bone marrow/solid organ transplantation, diabetes, chronic kidney disease and malnutrition).

### 2.4. Statistical Analysis

Data are described as median and interquartile range (IQR) or number and percentage. The population was divided into two groups according to survival at ICU discharge. Categorical variables were compared using a chi-square test and continuous variables using the nonparametric Wilcoxon test. ICU and 90-day mortalities of our studied population were compared to a control group of critically ill patients with aspergillosis. Survival curves were generated using the Kaplan–Meier methodology (compared using a log-rank test). All tests were two-sided, and *p*-values less than 0.05 were considered statistically significant. Analyses were done using R software version 4.0.2 (Free Software Foundation, Boston, MA, USA).

## 3. Results

### 3.1. Characteristics of the Study Population

During the study period, a total of 26 cases were identified and confirmed from our database. Given that 6900 patients were admitted in our ICU concomitantly, incidence of EIFI was at 4 per 1000 admissions (Figure 1).

Characteristics of the study population are summarized in Table 1. Patients were mainly male (81%) with a median age of 59 (48–69) years old. All of them had one or more predisposing factors for EIFI, including drug-induced immunodepression (85%, induced by corticosteroids or immunosuppressive drugs), hematological malignancies, transplantation and malnutrition (81%). A majority of patients were previously treated with broad-spectrum antibiotics (96%), had a central venous access (89%) and were colonized with *Candida* (85%).

Most of the patients had been admitted to the ICU for a septic shock (89%). At admission, the median SOFA score was at 10 (8–13) and the median SAPS II at 59 (49–76). Twenty-four patients (92%) had an AKI stage ≥ 1, 16 (62%) had an ARDS and 14 (54%) had a myocardial dysfunction, underlying the severity of illnesses. Life support therapies were therefore undertaken, at least one in each patient: invasive mechanical ventilation in 25, vasopressors in 23 and renal replacement therapy in 18 patients.

### 3.2. Mold Isolates

EIFI was proven in 18 patients (69%), probable in 5 patients (19%) and possible in 3 patients (12%). Median time elapsed from ICU admission to EIFI diagnosis was 9 (3–14) days. At EIFI diagnosis, median SOFA score increased to 13 (9–16). Bloodstream EIFI occurred in less than one-third of cases (31%), but EIFIs mainly involved lungs (54%) and skin (35%). *Mucorales* accounted for half of molds isolated. Other filamentous fungi were mainly *Saprochaete* (formally, *Geotrichum*), *Fusarium* and *Scedosporium* (Table 2).

### 3.3. Treatments

An antimicrobial susceptibility testing of fungi was available in only 7 patients (27%). The antifungal susceptibilities of isolates are reported in Table 3.

Though an antifungal therapy had been administered in most of the patients (92%), it was inadequate in 4 of them. The antifungal treatment was started within 24 h before and after EIFI diagnosis in more than 75% of patients. More than two-thirds of antifungal regimens included amphotericin B liposomal (69%), associated in 20% of cases with echinocandin, triazole or flucytosine. A surgical source control was performed in 5 patients (19%). It is noteworthy that 3 patients did not received any antifungal treatment, and EIFI diagnosis was confirmed after death (Table 2). Moreover, 3 patients had also candidemia (*C albicans, C. glabrata, C. auris*), and 3 patients had aspergillosis (*A. fumigatus* and *A. flavus*).

### 3.4. Outcomes

Twenty patients died in the ICU (77%) and 2 more 90 days after discharge, which brought mortality to 85% (Kaplan–Meier curve is displayed in Appendix A). Of note, withdrawal of life sustaining treatment had been instituted in more than one-third (35%) of patients. By univariate analysis, prior conditions such as hematological malignancy or cancer (*p* = 0.041), chronic kidney disease (*p* = 0.025) and malnutrition (*p* = 0.047) were associated with in-ICU mortality. It was also associated with the development of ARDS (*p* = 0.025) and hepatic dysfunction (*p* = 0.044) during the ICU stay. By contrast, antifungal agent used, time duration to adequate antifungal therapy and surgical treatment were not significantly associated with mortality.

We first compared critically ill patients with mucormycosis to those with other EIFIs (namely, *Saprochaete*, *Fusarium*, *Scedosporium*, *Trichosporon*, *Chaetomium* and *Saccharomyces*), and we found that the former were significantly younger, mostly with rhino–cerebral and skin sites of infection, but with a similar mortality and severity (Appendix A and Appendix A).

We then compared our study population to a matched control group (age, severity and previous conditions) of critically ill patients with invasive aspergillosis (Table 4). The control group had mostly invasive pulmonary aspergillosis and received adequate antifungal treatment more frequently (*p* = 0.01). Nonetheless, day-90 survival probability was similar between patients with aspergillosis and those with EIFIs at 18%, 95% CI (8–45) and 20%, 95% CI (9–45), respectively (Figure 2).

## 4. Discussion

This study of 26 critically ill patients with EIFIs, representing a 14-year cohort, showed that the incidence of EIFIs remained low at 4 per 1000 admissions, but it was associated with high in-ICU and 90-day mortalities at 77% and 85%, respectively. We found also that main factors associated with mortality were (1) previous status including prior hematological malignancy or cancer, chronic kidney disease and malnutrition; and (2) the occurrence of an ARDS or a hepatic dysfunction during ICU stay. The comparison of this cohort with a group matched on age, prior conditions and severity with aspergillosis did not find any differences in mortality rates, suggesting that previous clinical status and severity of ICU stay may have influenced outcome more than the type of mold involved.

The incidence of EIFIs we observed was relatively high, particularly when compared to that of ICU-acquired candidemia (2 to 7 per 1000 admissions) or invasive aspergillosis [1]. However, these results must be qualified in light of the specificity of our patients, who are often immunocompromised and coming from onco-hematology wards, all well-known risk factors for mucormycosis and other emerging fungal infections [2,7,9,24]. Nonetheless, mortality rates were very high, reaching more than 75% at ICU discharge and 85% 90 days after ICU admission. Two recent multicentric studies of 74 and 26 critically ill patients with mucormycosis reported similar ICU mortality rates at 71.6% and 77%, respectively [7,25]. ICU outcomes of immunocompromised patients, especially those with ARDS [26] or/and hematological malignancies, are also close to our cohort [27].

We found that previous conditions such as malnutrition, hematological malignancies and chronic kidney diseases and the severity of events leading to ICU, mainly the occurrence of ARDS, were significantly associated with mortality. Whether the occurrence of an EIFI worsen the prognosis or is just a marker of severity is questionable [28]. The comparison of our cohort study to a matched (same previous clinical status and severity) control group with aspergillosis showed that mortality did not differ irrespective of mold species identified. It means that levels of immunodepression and malnutrition and the severity of organ failure may be the main determinants of EIFI occurrence and worse prognosis [25,28]. By contrast, time duration to adequate antifungal therapy, type of antifungal agent and surgical management were not associated with mortality. It has been reported that surgical debridement [7,29,30] and early antifungal treatment [31] may improve outcome in patients with mucormycosis [2,9,22,24,32,33,34,35]. The use of combination antifungal therapy did not seem to impact the survival of our patients [36]. However, these results should be taken with caution given the number of patients analyzed and the fact that surgical management was mostly dependent on clinical status. Large cohort studies concluded that only the fungal infection site influenced significantly outcomes in patients with mucormycosis [24]. Earlier diagnosis and immune reconstitution are probably the unmet needs to improve outcomes. Future development axes to improve the prognosis would involve new laboratory diagnostic and imaging procedures and the search for new antifungal compounds. In any case, drug monitoring should be a preliminary to any antifungal treatment because a large proportion of patients exhibit low drug concentrations leading to poor outcomes, especially in ICU settings [37].

We must acknowledge some limitations to our study. Our conclusions are limited by the relatively small size of the cohort and by the retrospective design of the study, which could induce bias in data collection and results interpretation. Particularly, our estimation of EIFI incidence may have been flawed due to the small number of included patients. However, estimation of incidence of rare diseases remains difficult. It is noteworthy that the incidence of EIFIs in our study is within the same range of EIFI incidence previously reported [1,38,39]. Moreover, to the best of our knowledge, this is the first study that aimed to assess epidemiological trends, clinical features and outcomes of EIFIs other than mucormycosis in ICU settings. Lastly, our results should not be generalized to all ICUs because of the specificity of ours or to other countries due to the variability in geographical distribution of fungal infections.

## 5. Conclusions

In this 14-year cohort study, the incidence of EIFIs in critically ill patients, far from negligible, has been estimated at 4 per 1000 admissions, leading to very poor outcomes. Main factors associated with mortality were prior conditions including hematological malignancy or cancer, malnutrition and chronic kidney disease as well as severe pulmonary involvement. It would appear that the state of immunosuppression, the clinical status and the severity of the acute event are more determinant in the prognosis than the type of fungal infection. Giving that the precise evaluation of fungal strain weight on patients’ outcomes is complex, our results may be taken with caution. Outcomes may depend also on the fungus’ affinity to tissues, its virulence factors and its intrinsic antimycotic resistance. Nonetheless, further studies are mandatory to estimate the incidence of such diseases and to assess the effects of an earlier diagnosis and appropriate treatment on the outcome.

## Figures and Tables

**Figure 1 jof-07-00330-f001:**
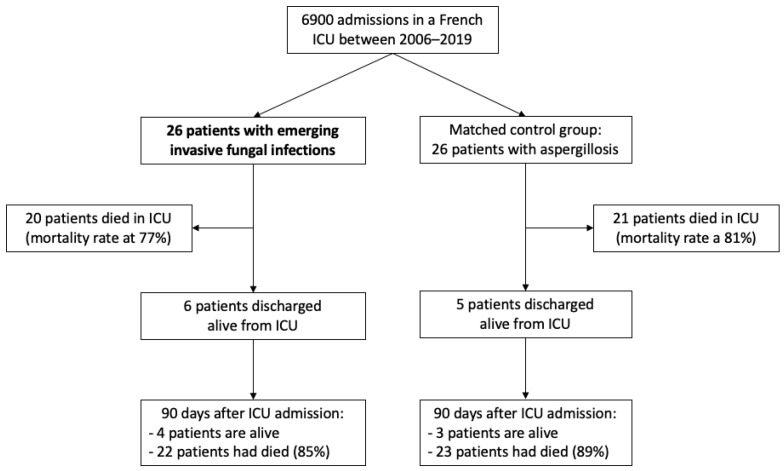
Flow chart of the study population.

**Figure 2 jof-07-00330-f002:**
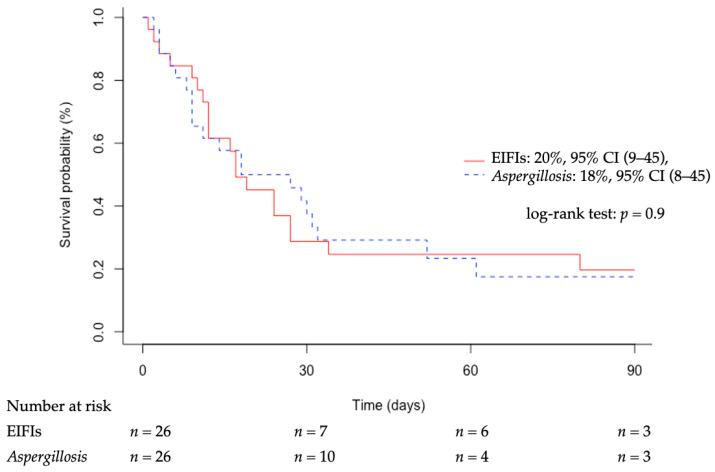
Kaplan–Meier curve of day-90 survival after ICU admission of 26 patients with emerging invasive fungal infections (red line) and 26 patients with aspergillosis (blue dashed line). Emerging invasive fungal infections (EIFIs): *Mucorales* (*n* = 13), *Saprochaete* (*n* = 6), *Trichosporon* (*n* = 1), *Fusarium* (*n* = 2), *Scedosporium* (*n* = 2), *Chaetomium* (*n* = 1), *Saccharomyces* (*n* = 1). 95% CI: 95% confidence interval.

**Table 1 jof-07-00330-t001:** Characteristics of the study population and univariate analysis for in-ICU mortality.

	Total (*n* = 26)	Survivor (*n* = 6)	Deceased (*n* = 20)	*p*-Value
Male, n (%)	21 (81%)	4 (67%)	17 (85%)	0.33
Age (years), median (IQR)	58.5 (48–69)	63.5 (52–69)	57 (47–68)	0.81
Charlson index, median (IQR)	3 (2–5)	1 (0–4)	3 (2–4)	0.2
BMT/SOT, n (%)	14 (54%)	2 (33%)	12 (60%)	0.26
Hematological mal./cancer, n (%)	15 (58%)	1 (17%)	14 (70%)	0.04
Immunosuppressors, n (%)	22 (85%)	4 (67%)	18 (90%)	0.19
Corticosteroids, n (%)	18 (69%)	4 (67%)	14 (70%)	0.88
Diabetes, n (%)	6 (23%)	1 (17%)	5 (25%)	0.67
CKD, n (%)	4 (15%)	3 (50%)	1 (5%)	0.03
Burn/Trauma, n (%)	3 (12%)	2 (33%)	1 (5%)	0.09
Malnutrition, n (%)	21 (81%)	3 (50%)	18 (90%)	0.047
*Candida* colonization, n (%)	22 (85%)	5 (83%)	17 (85%)	0.92
Central venous catheter, n (%)	23 (89%)	5 (83%)	18 (90%)	0.66
Broad-spectrum antibiotic, n (%)	25 (96%)	6 (100%)	19 (95%)	>0.99
SAPS II, median (IQR)	59 (49–76)	65 (59–78)	57 (48–68)	0.48
SOFA at admission, median (IQR)	10 (8–13)	10 (8–11)	11 (8–13)	0.79
Invasive mechanical ventilation, n (%)	25 (96%)	5 (83%)	20 (100%)	0.99
Duration (days), median (IQR)	11 (4–24)	30 (10–38)	11 (5–21)	0.13
ARDS, n (%)	16 (62%)	1 (17%)	15 (75%)	0.03
Vasoconstrictive drugs, n (%)	23 (89%)	5 (83%)	18 (90%)	0.66
Duration (days), median (IQR)	13 (10–28)	30 (29–30)	12 (9–17)	0.2
AKI, n (%)	24 (92%)	5 (83%)	19 (95%)	0.37
RRT, n (%)	18 (69%)	4 (67%)	14 (70%)	0.88
Duration (days), median (IQR)	10 (6–25)	45 (31–63)	9 (4–18)	0.06
Hepatic dysfunction, n (%)	18 (69%)	2 (33%)	16 (80%)	0.04
Myocardial dysfunction, n (%)	14 (54%)	1 (17%)	13 (65%)	0.06
Mucormycosis, n (%)	13 (50%)	3 (50%)	10 (50%)	>0.99
Other EIFIs, n (%)	13 (50%)	3 (50%)	10 (50%)	>0.99
Organ involvements:				
Pulmonary, n (%)	14 (54%)	1 (17%)	13 (65%)	0.06
Cutaneous, n (%)	9 (35%)	2 (33%)	7 (35%)	0.94
Blood, n (%)	8 (31%)	3 (50%)	5 (25%)	0.26
Sinus/orbit, n (%)	4 (15%)	0 (0%)	4 (20%)	>0.99
Cerebral, n (%)	2 (8%)	0 (0%)	2 (10%)	>0.99
Antifungal therapy	24 (92%)	6 (100%)	18 (90%)	>0.99
Surgery, n (%)	5 (19%)	2 (33%)	3 (15%)	0.33
ICU LOS (days), median (IQR)	17 (9–27)	46 (15–51)	14 (10–24)	0.15
Deceased in ICU, n (%)	20 (77%)	0 (0%)	20 (100%)	>0.99
Deceased at day-90, n (%)	22 (85%)	2 (33%)	20 (100%)	>0.99
WLST, n (%)	9 (35%)	2 (33%)	7 (35%)	0.94

BMT: bone marrow transplantation (*n* = 12), SOT: solid organ transplantation (kidney, *n* = 2), Hematological mal.: hematological malignancies (*n* = 14), CKD: chronic kidney disease, LOS: length of stay, ICU: intensive care unit, SAPS II: Simplified Acute Physiology Score II, SOFA: Sequential Organ Failure Assessment, ARDS: acute respiratory distress syndrome, AKI: acute kidney injury (KDIGO stage ≥ 1), RRT: renal replacement therapy, EIFI: emerging invasive fungal infection, WLST: withdrawal of life sustaining treatment.

**Table 2 jof-07-00330-t002:** Details on molds isolated and antifungal treatments.

	Total (*n* = 26)	Survivor (*n* = 6)	Deceased (*n* = 20)	*p*-Value
Mucormycosis, n (%)	13 (50%)	3 (50%)	10 (50%)	>0.99
*Rhizopus* sp., n (%)	9 (35%)	1 (17%)	8 (40%)	0.31
*Lichtheimia corymbifera*, n (%)	2 (8%)	2 (33%)	0 (0%)	>0.99
*Rhizomucor pusillus*, n (%)	1 (4%)	0 (0%)	1 (5%)	0.99
*Mucor circinelloides*, n (%)	1 (4%)	0 (0%)	1 (5%)	0.99
Other EIFIs, n (%)	13 (50%)	3 (50%)	10 (50%)	>0.99
*Saprochaete clavata*, n (%)	1 (4%)	1 (17%)	1 (5%)	0.59
*Saprochaete capitata*, n (%)	4 (19%)	0 (0%)	4 (20%)	0.99
*Trichosporon asahii*, n (%)	1 (4%)	0 (0%)	1 (5%)	0.99
*Fusarium* sp., n (%)	2 (8%)	0 (0%)	2 (10%)	>0.99
*Scedosporium apiospermum*, n (%)	2 (8%)	0 (0%)	2 (10%)	>0.99
*Chaetomium cymbiformis*, n (%)	1 (4%)	1 (17%)	0 (0%)	0.99
*Saccharomyces cerevisiae*, n (%)	1 (4%)	1 (17%)	0 (0%)	0.99
Antifungal therapy	24 (92%)	6 (100%)	18 (90%)	>0.99
Adequate	20 (77%)	5 (83%)	15 (75%)	0.67
Amphotericin B, n (%)	18 (69%)	3 (50%)	15 (75%)	0.26
Echinocandin, n (%)	6 (23%)	1 (17%)	5 (25%)	0.67
Fluconazole, n (%)	1 (4%)	1 (17%)	0 (0%)	0.99
Voriconazole, n (%)	2 (8%)	0 (0%)	2 (10%)	>0.99
Isavuconazole, n (%)	2 (8%)	2 (33%)	0 (0%)	>0.99
Flucytosine, n (%)	1 (4%)	1 (17%)	0 (0%)	0.99
Association, n (%)	5 (19%)	2 (33%)	4 (20%)	0.37

Among *Rhizopus* sp.: *Rhizopus oryzae* (*n* = 3), *Rhizopus microsporus* (*n* = 1). *Lichtheimia corymbifera* is formerly known as Absidia corymbifera or *Mycocladus corymbifer*, *Saprochaete clavata* is formerly known as *Geotrichum clavatum* and *Saprochaete capitata* is formerly known as *Geotrichum capitatum* or *Trichosporon capitatum*.

**Table 3 jof-07-00330-t003:** Antifungal therapy susceptibility testing of emerging molds in the study population.

	Amphotericin B MIC ^1^ (mg/L)	Caspofungin MIC ^1^ (mg/L)	Fluconazole MIC ^1^ (mg/L)	Voriconazole MIC ^1^ (mg/L)	Posaconazole MIC ^1^ (mg/L)	Isavuconazole MIC ^1^ (mg/L)	Flucytosine MIC ^1^ (mg/L)
*S. clavata*	1–1.5	32	8	0.5		0.5	0.25
*S. capitata*	0.5–1	32–32	0.19–24	0.004–0.19	0.016–0.38	*	*
*T. asahii*	2	32	0.75	0.023			
*S. cerevisiae*	*	0.5	3	*	*	*	*

^1^ Minimal inhibitory concentrations (MICs) were available in 1 strain of *Trichosporon asahii*, 1 strain of *Saccharomyces cerevisiae* and in 2 strains of *Saprochaete clavata* (formally, *Geotrichum clavatum*) and 3 strains of *Saprochaete capitata* (formally, *Geotrichum capitatum* or *Trichosporon capitatum*) in which a range of MIC were reported. Interpretative breakpoints were not available for most drugs or fungi, and a MIC < 1 mg/L was generally used as an indicator of susceptibility for all drugs except for fluconazole, MIC < 2–4 mg/L and flucytosine, MIC < 1–16 mg/L [22,23]. * unavailable data, susceptibility has not been tested.

**Table 4 jof-07-00330-t004:** Comparison of the EIFI cohort to a matched control group with aspergillosis.

	EIFIs (*n* = 26)	Aspergillosis (*n* = 26)	*p*-Value
Age (years), median (IQR)	59 (48–69)	57 (44–64)	0.29
Charlson index, median (IQR)	3 (2–5)	3 (2–4)	0.39
Malnutrition, n (%)	21 (81%)	20 (77%)	0.74
Hematological mal/cancer, n (%)	15 (58%)	16 (62%)	0.78
Diabetes, n (%)	6 (23%)	4 (15%)	0.49
CKD, n (%)	4 (15%)	4 (15%)	>0.99
SAPS II, median (IQR)	59 (49–76)	51 (42–60)	0.15
SOFA at admission, median (IQR)	10 (8–13)	11 (8–13)	0.56
ARDS, n (%)	16 (62%)	16 (62%)	>0.99
Deceased in ICU, n (%)	20 (77%)	21 (81%)	0.74
Deceased at day-90, n (%)	22 (85%)	23 (89%)	0.64

CKD: chronic kidney disease, ICU: intensive care unit, SAPS II: simplified acute physiology score II, SOFA: sequential organ failure assessment, ARDS: acute respiratory distress syndrome, EIFI: emerging invasive fungal infection.

## Data Availability

The authors consent to share the collected data with others. Data will be provided to qualified investigators free of charge after careful examination of required documents (summary of the research plan, request form and IRB approval) by the study board of investigators. Data will be available immediately after the main publication and indefinitely.

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
