# Peer review of "Emerging Invasive Fungal Infections in Critically Ill Patients: Incidence, Outcomes and Prognosis Factors, a Case-Control Study"

_jof, 2021, doi:10.3390/jof7050330_

Round 1
Reviewer 1 Report
Dear Authors,
The manuscript ID: jof-1164835 entitled “Emerging invasive fungal infections in critically ill patients: incidence, outcomes and prognosis factors, a case-control study” written by Romaric Larcher, Laura Platon, Matthieu Amalric, Vincent Brunot, Noemie Besnard, Racim Benomar, Delphine Daubin, Patrice Ceballos, Philippe Rispail, Laurence Lachaud, Nathalie Bourgeois and Kada Klouche is very interesting.
Authors described in this retrospective critically ills cohort study, the epidemiological trends, clinical features and outcome of emerging invasive fungal infections and to assess prognostic factors in intensive care unit settings. In this work was also compared cohort of intensive care unit patients with these invasive infections to a matched control group with aspergillosis in order to evaluate the weight of these species on outcome.
This manuscript is good documented and appropriately organized. Hovewer, in my opinion, the study cohort size was too small and the results are not sufficient to be accepted. I think, this article is not worth publishing in such a prestigious journal as "Journal of Fungi".
With highest regards,

Author Response
Dear Authors,
The manuscript ID: jof-1164835 entitled “Emerging invasive fungal infections in critically ill patients: incidence, outcomes and prognosis factors, a case-control study” written by Romaric Larcher, Laura Platon, Matthieu Amalric, Vincent Brunot, Noemie Besnard, Racim Benomar, Delphine Daubin, Patrice Ceballos, Philippe Rispail, Laurence Lachaud, Nathalie Bourgeois and Kada Klouche is very interesting.
Authors described in this retrospective critically ills cohort study, the epidemiological trends, clinical features and outcome of emerging invasive fungal infections and to assess prognostic factors in intensive care unit settings. In this work was also compared cohort of intensive care unit patients with these invasive infections to a matched control group with aspergillosis in order to evaluate the weight of these species on outcome.
This manuscript is good documented and appropriately organized. Hovewer, in my opinion, the study cohort size was too small and the results are not sufficient to be accepted. I think, this article is not worth publishing in such a prestigious journal as "Journal of Fungi".
With highest regards,
First, we thank the reviewer for his/her interest to this work.
We agree with the reviewer that the cohort size may appear small and we have stated this in the limit section of discussion (please see limit section of discussion, lines 684-686, p8).
We acknowledge however, that our study is the first to report the outcomes of other emerging fungal infections than mucormycosis in ICU patients and to estimate the weight of fungal species on prognosis. Moreover, a recent cohort study of 26 ICU patients with mucormycosis reported by Jestin et al. has been recently published in Annals of Intensive Care which is also a prestigious journal (IF 4.304, Q1).
We hope that our results, despite the small size of our cohort, may add substantial information and help intensivists and other physicians to better manage patients with these rare and severe diseases.
Reviewer 2 Report
This is a good paper and findings are generally well described. I have only a few observations:
Line 94: The authors have cited references 9 and 17-20 in relation to auditing the management. However, their data precedes the year of these citations. What could be considered inadequate now might have been adequate in real time. This deficiency needs to be mentioned clearly.
Line 97: How were the controls selected?
Line 142: Geotrimcum - it is a typo. Please correct
Table 2: What antifungals in which situations were considered inadequate? It appears that 20 out of 23 patients who received therapy were given adequate therapy.
Figure 1: The representation has scope for improvement.
References are not cited in a consistent manner. At times the journal name is in upper case (ref 30), at times in full (ref 20), and at times abbreviated (ref 23).
Author Response
This is a good paper and findings are generally well described.
The authors thank the reviewer for being interested in this work and for his/her remarks.
I have only a few observations:
Line 94: The authors have cited references 9 and 17-20 in relation to auditing the management. However, their data precedes the year of these citations. What could be considered inadequate now might have been adequate in real time. This deficiency needs to be mentioned clearly.
Adequation of antifungal therapies were retrospectively confirmed by two investigators (R. L., and L.P.), in agreement with the latest data/guidelines. Indeed, some treatments have been considered inadequate whereas they were considered adequate at treatment start date. The method section has been amended accordingly, please see lines 145-149, p3.
Line 97: How were the controls selected?
Controls were selected among patients admitted to our ICU during the same 14-year study period with a diagnosis of aspergillosis according to Infectious Diseases Society of America (IDSA) guidelines. Controls were then matched on the basis of age (+/- 2 years old), SOFA score (+/- 1), development of ARDS, and previous conditions (at least 3 similar conditions among: hemopathy/cancer, bone marrow/solid organ transplantation, dia-betes, chronic kidney disease and malnutrition). We amended the text accordingly, please see Method section, lines 153-158, p3.
Line 142: Geotrimcum - it is a typo. Please correct
Done. Please see line 360, p5
Table 2: What antifungals in which situations were considered inadequate? It appears that 20 out of 23 patients who received therapy were given adequate therapy.
Indeed, 20 out of 24 patients who received antifungals received adequate therapy. An inadequate antifungal treatment was noticed in two patients with Mucorales, one patient with fusariosis and the last with scedosporosis.
One patient get an infected leg wound and had an empiric treatment with fluconazole. An Absidia corymbifera has been documented. However, a surgical amputation of the leg was performed before documentation and the patient was discharge alive from ICU without specific antifungal treatment.
A patient with radiologic pulmonary infiltrates and a cerebral abscess has been empirically treated with voriconazole. He died before Mucoracee was documented.
Lastly, 2 patients had an empirical treatment with caspofungin and deceased before the finding of fusariosis and scedosporosis, respectively.
Figure 1: The representation has scope for improvement.
The figure 1 has been entirely rebuild. Please see figure 1.
References are not cited in a consistent manner. At times the journal name is in upper case (ref 30), at times in full (ref 20), and at times abbreviated (ref 23).
All references have been cited according to the Journal of Fungi guidelines (Author 1, A.B.; Author 2, C.D. Title of the article. Abbreviated Journal Name Year, Volume, page range.). Please see references pp9-12.
Reviewer 3 Report
Introduction: add short statement on other emerging fungal pathogens as well.
Line 142: correctly Geotrichum.
Legend Table 2: fungal names in Italics; correctly "formerly known".
Table 1 and 3: do not leave empty fields.
Table 3: correctly Saprochaete clavata.
Line 192: correctly 14-year.
Conclusion: The precise role of the specific fungal strain infection in patient´s prognosis/outcome must be evaluated also in relationship to the fungus affinity to specific tissue/tissues, to its particular virulence (factors), to its intrinsic antimycotic resistance, antiimmune mechanisms etc. - huge complexity.
Author Response
Introduction: add short statement on other emerging fungal pathogens as well.
We added a short statement on other EIFI in the introduction, please see lines 98-100, p 2.
Line 142: correctly Geotrichum.
Done. Please see line 360, p5
Legend Table 2: fungal names in Italics; correctly "formerly known".
Done. Please see line 361, p6
Table 1 and 3: do not leave empty fields.
Done. Please see table 1 and table 3.
Table 3: correctly Saprochaete clavata.
Done. Please see lines 553-555, p6
Line 192: correctly 14-year.
Done. Please see line 643, p8
Conclusion: The precise role of the specific fungal strain infection in patient´s prognosis/outcome must be evaluated also in relationship to the fungus affinity to specific tissue/tissues, to its particular virulence (factors), to its intrinsic antimycotic resistance, antiimmune mechanisms etc. - huge complexity.
As correctly pointed by the reviewer it will be interesting to evaluate the weight of fungal strain on outcome taking into account the fungus affinity to tissues, to its virulence factors and its intrinsic antimycotic resistance. This complexity limits the interpretation of our results, and we therefore added a statement in the conclusion. Please see lines 698-701, p9.
The authors thank the reviewer for being interested in this work and for thorough comments.
Reviewer 4 Report
The manuscript describes the results of a retrospective observational cohort study extended on 14 years. It is of great interest for infection disease specialists and also for specialists dealing with hematological patients. Generally speaking, the paper is well written. Only few minor corrections are necessary.
Lines 74-75: A short presentation of the criteria should be included.
Line 192: replace "a thirteen-year cohort" with "a fourteen-year cohort"
Author Response
The manuscript describes the results of a retrospective observational cohort study extended on 14 years. It is of great interest for infection disease specialists and also for specialists dealing with hematological patients. Generally speaking, the paper is well written. Only few minor corrections are necessary.
First, we thank the reviewer for his/her interest to this work and for his helpful observations.
Lines 74-75: A short presentation of the criteria should be included.
As suggested by the reviewer, we added a presentation of the EIFI diagnosis criteria in the method section. Please see lines 122-126, p2.
Line 192: replace "a thirteen-year cohort" with "a fourteen-year cohort"
Done. Please see line 643, p8
Round 2
Reviewer 1 Report
Dear Authors,
I have read the revised version of the manuscript ID: jof-1164835 entitled “Emerging invasive fungal infections in critically ill patients: incidence, outcomes and prognosis factors, a case-control study”. I agree with the Authors that the study is the first to report the outcomes of other emerging fungal infections than mucormycosis in invasive fungal infections patients. Indeed, these results may add important information and help physicians to better manage patients with these diseases. The Authors have made also corrections to this paper, as suggested by all reviewers. I am recommending your manuscript for publication in “Journal of Fungi”.
With highest regards,
